# Using mobile audiometry (Wulira App) to assess noise induced hearing loss among industrial workers in Kampala, Uganda: A cross-sectional study

**Charles Batte**[1]*, **Immaculate Atukunda**[2], **Andrew Weil Semulimi**[1],
**Mariam Nakabuye**[1], **Festo Bwambale**[3], **Joab Mumbere**[4], **Nelson Twinamasiko**[1],
**David Mukunya**[5], **Israel Paul Nyarubeli**[6], **John Mukisa**[7]

1 Lung Institute, Department of Medicine, School of Medicine, College of Health Sciences, Makerere University, Kampala, Uganda, 2 Department of Ophthalmology, School of Medicine, College of Health Sciences, Makerere University, Kampala, Uganda, 3 Department of Clinical Medicine, Mulago National Referral Hospital, Kampala, Uganda, 4 Department of Information, Technology and Communications, Wulira Health Limited, Kampala, Uganda, 5 Department of Community and Public Health, Faculty of Health Sciences, Busitema University, Mbale, Uganda, 6 School of Public Health and Social Sciences, Muhimbili University of Health and Allied Sciences (MUHAS), Dar es Salaam, Tanzania, 7 Department of Immunology and Molecular Biology, School of Biomedical Sciences, College of Health Sciences, Makerere University, Kampala, Uganda

☯ These authors contributed equally to this work.
* dr.cbatte@gmail.com

## Abstract

### Background

Occupational noise is a common cause of hearing loss in low-income countries. Unfortunately, screening for hearing loss is rarely done due to technical and logistical challenges associated with pure tone audiometry. *Wulira* app is a valid and potentially cost-effective alternative to pure tone audiometry in screening for occupational hearing loss. We aimed to determine the prevalence of occupational hearing loss among workers in a metal industry company in Kampala district.

### Methodology

We recruited 354 participants conveniently from a steel and iron manufacturing industry in Kampala. All eligible participants answered a pretested and validated questionnaire and were assessed for noise induced hearing loss in a quiet office room approximately 500 meters from the heavy machinery area using the Wulira app. Descriptive statistics such as proportions were used to describe the study population while inferential statistics were used to determine associations.

### Results

Of the 354 participants sampled, 333 (94.1%) were male, and the median age was 27, IQR (25–30). Regarding the risk factors of hearing loss, fourteen (3.9%) had history of smoking and more than half (65.5%) had worked in the industry for more than 2 years. The overall

**Data Availability Statement:** All the scientific data used in analysis is available from figshare database (10.6084/m9.figshare.20013560).

**Funding:** This study was funded by the Government of Uganda through the Research and Innovation Fund Makerere University, Fund MAKRIF/ DVCFA/ 026/ 20. AWS and MN are research fellows of the MakNCD program supported by the Fogarty International Centre of the National Institutes of Health under Award Number D43TW011401. The content is solely the responsibility of the authors and does not necessarily represent the official views of the Research and Innovation Fund or the National Institutes of Health.

**Competing interests:** The authors have read the journal's policy and have the following competing interests: CB is part of the team that developed the Wulira App. CB, JM, and AWS serve as employees of Wulira Health Limited in which Wulira App is a subsidiary. This does not alter our adherence to PLOS ONE policies on sharing data and materials. The other authors have no conflict of interest to declare.

prevalence of hearing loss among industrial workers was 11.3% (40/354). 16.2% and 9% had mild hearing loss in the right and left ear respectively. Bilateral audiometric notch was present where fourteen (4%) of the participants had notch in their right ear while seven (2%) had notch in their left ear. Residing outside Kampala district was associated with hearing loss (OR, 95% CI, 0.213 (0.063–0.725), p = 0.013).

## Conclusion

One in 10 workers in a metal manufacturing industry in Kampala had occupational hearing loss. Industrial workers residing outside Kampala were likely to develop hearing loss. Periodic screening should be done for early detection and intervention to prevent progression of hearing loss in this population.

## Introduction

The World Health Organisation (WHO) reports that close to half a billion people suffer from disabling hearing loss which is expected to rise to 630 million by 2030 [1]. This is partly due to the increased level of mechanisation in industries resulting into the production of harmful excessive harmful noise [2] which has predisposed industrial workers to sensorineural hearing loss [3, 4]. As a result of the industrial revolution [4], Occupational Hearing Loss (OHL) is increasing becoming a serious public health problem in Sub-Saharan Africa (SSA). Epidemiological studies have shown that the prevalence of OHL ranges between 17% and 48% [3, 5–8]. In Uganda, over 11% of the working population has hearing difficulty, and nearly one out of four cases of OHL are caused by several exposures which include noise, fumes, and heat [3, 4, 9–12]. This presents a significant hindrance to Uganda's efforts of achieving middle income status and the third United Nations Sustainable Development Goal (SDGs) which is to achieve good health and wellbeing.

In Uganda, metal manufacturing industries contribute significantly to the economy of the country through provision of employment and payment of taxes [13–15]. However, studies have reported that metal manufacturing industries produce high noise levels ranging from 90.5–105 dB (A) which is above the recommended noise level 75 to 85 dB (A) [9, 16–20] increasing the risk of developing OHL among the workers [21]. OHL ranks among the leading causes of occupational illness among industrial workers [3, 6] and is responsible for 46 million years lived with disability (YLD) [22]. It has devastating effects on industrial workers such as social isolation, impaired communication with co-workers and family, decreased ability to monitor the work environments, increased injuries, lost productivity, anxiety [23, 24], and an increased risk of cardiovascular diseases/ hypertension [25, 26]. The absence of hearing protection devices in these industries [17] further exacerbate the risk.

To reduce the risk of OHL, early detection and prevention is critical in addressing this occupational hazard. To differentiate age related hearing loss from OHL, the use of audiometric notch has been proposed [27]. This can be used to make a diagnosis of OHL [28] and identify genetic risk to hearing loss [29]. Much as the Occupational and Health safety Act 2006 requires employers to provide periodic screening for OHL for workers in environments that expose them to harmful noise, majority of the employers have failed to implement this law. This is because Pure Tone Audiometry (PTA), which is the gold standard for hearing loss screening, is quite costly, requires a specialist audiologist to operate in a sound- proof room and is not readily available at health sites in Sub-Saharan Africa.

However, mobile audiometry, which is hearing loss screening using mobile phones and tablets, could bridge this challenge. One of the available mobile phone -based hearing loss screening tools is the Wulira app which has a specificity of 93.2% and 91.5% for the right and left ear respectively and sensitivity of 91.4% and 88.4% for the right and left ear [30]. This shows that the Wulira App can be used as an alternative of PTA to screen for hearing loss especially in settings with limited access to PTA [30–32]. Mobile audiometry has been utilized in the general population to screen for hearing loss [30–32] providing evidence for its use in industrial workers and to guide formulation of policies aimed at improving occupational health and safety. We hypothesized that industrial workers at a steel-manufacturing factory in Uganda, have a high prevalence of noise induced hearing loss. The purpose of our study was to determine the prevalence of noise induced hearing loss among industrial workers using the Wulira App and the factors associated with it. We hypothesized that there would be no differences in the presence of audiometric notch among the study participants when the duration of working in the factory were considered.

## Methods

### Study design and site

We conducted a cross-sectional study among participants from a metal manufacturing industry in Kampala district from January 2021 to April 2021. We approached all the three potential metal manufacturing industries in Kampala who had more than 400 employees. Only one factory granted us permission to recruit their employees and a non-disclosure agreement was signed with the industry.

### Study population

The industry employs over 900 workers of which an estimated 400 individuals work with machines that produce noise. Individuals with hearing loss are excluded from working at the factory during the pre-employment hearing evaluation. We retrieved a list of employees who worked in the noisy sections of the industry from the management and randomly selected 354 participants. There was no measure of ambient noise in the factory sections. Study participants who worked more than 4 hours a day in the highly noisy machine areas were approached to participate in the study. Industrial workers who were 18 years and above and had consented to take part were included into the study. We excluded factory employees who used hearing aids, had current history of bilateral outer, middle, or inner ear infectious pathology based on examination by an audiologist. Those with vertigo were considered to have benign paroxysmal positional vertigo, which minimally affected their participation in the hearing assessments.

### Sample size and recruitment

Using the Kish and Leslie formula for a single proportion [33], at a desired precision of 0.05, design effect of 2.0, non-response rate of 10%, and assuming the prevalence of hearing loss from the general population was 11.7% [12] (due to absence of previous studies on OHL in Uganda), a sample size of 354 participants was determined. Participants were recruited from sections of the industry that were deemed noisy by the management of the industry.

### Data collection procedures

The participants answered a pre-tested interviewer guided questionnaire which collected information on their socio-demographic and clinical history. This included data on age, how long they had worked in the industry, previous history of smoking, and multi-drug resistant tuberculosis (MDR-TB) treatment. We collected data on MDR-TB treatment because of the

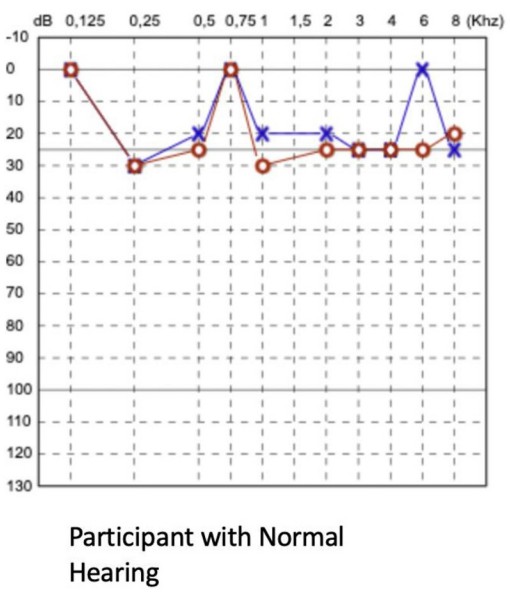

Participant with Normal
Hearing

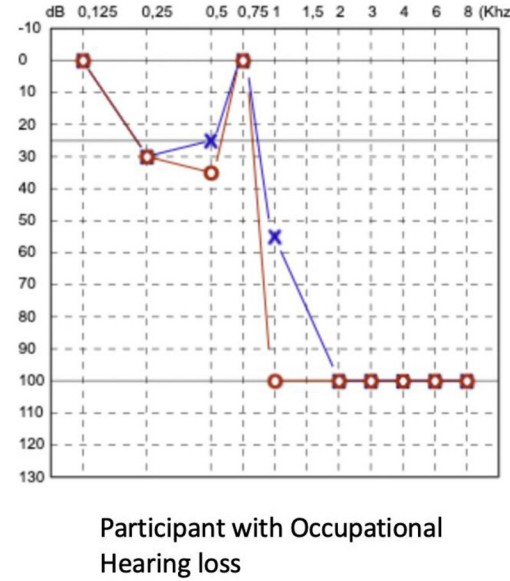

Participant with Occupational
Hearing loss

**Fig 1. Audiograms generated from Wulira App.**

effect of aminoglycosides on the auditory apparatus. An electronic data collection tool, Kobo-Toolbox [34], was used to collect data. Then, the participants undertook hearing loss screening test using the Wulira App administered by the audiologist in a quiet room approximately 500 meters from the heavy machinery area. The heavy machinery area has noise absorbing boards to minimize the noise transmission outside the machine housing unit of the factory. The Wulira App generated audiograms (Fig 1) which were interpreted by a trained audiologist. Noise cancelling headphones, Sennheiser HD 280 Pro, ISO 9001, Sennheiser electronic GmbH & Co. KG, Wedemark, Germany [35], were used in the administration of the test. Frequencies (250–8000 Hz) were sent to the participant ears following a manual test procedure consistent with international standards (1SO 8253–1:2010) [36, 37].

OHL was defined as an average hearing threshold more than or equal to 26 dB. It was graded based on the WHO classification of hearing loss, slight/ mild cut off 26–40 dB, moderate 41–60 dB, severe 61–80 dB, and profound being greater than 81 dB [38]. Audiometric notch was defined when the 4000 Hz threshold minus the 2000 Hz threshold and the 4000 Hz threshold minus the 8000 Hz threshold both were ≥10 dB [39].

## Quality control and assurance

We pretested questionnaires and modified the questionnaires to improve their consistence. The questionnaires were pretested among 10 non-industrial workers who were not part of the study population. Research assistants were trained in Good Clinical Practice (GCP). All hearing tests were conducted by a trained audiologist using the Wulira App. All entered data was validated using data cleaning codes. Hearing tests results were reviewed for completeness and linked with the data from the questionnaires.

## Data analysis

Data analysis was done using Stata V14. In the initial steps of the analysis, we described the study population using the exposure variables like duration of work in the factory, age, sex,

smoking, previous MDR-TB treatment. This involved presenting the frequencies and percentages for categorical data; means and standard deviations, medians, and interquartile ranges for the continuous variable. A Graph was used to describe the difference between subjective and objective OHL. A pie chart was used to describe the roles of the workers in the industry. Considering the OHL as the dependent variable, we determined the prevalence of OHL as a proportion of workers who had hearing loss based on the Wulira App. We compared the audiometric notch presence with the duration the participants had worked in the factory using the Fisher's exact test. A p-value of less than or equal to 0.05 was considered as statistically significant. A logistic regression analysis was performed to identify any associations between demographics, work history and hearing loss (as the outcome). Confounding was assessed by considering a 15% change in the odds ratio when a model with the variable and one without. Two-way interaction terms were and the significance of the association assessed using the likelihood ratio test. The odds ratios and their 95% confidence intervals were presented.

## Ethics

Ethical Approval was obtained from Makerere University School of Biomedical Sciences Institutional Review Board (SBS-862) and Uganda National Council of Science and Technology (HS1237ES) before data collection. Administrative clearance was sought from the head of the industry and a non-disclosure agreement signed to maintain confidentiality. Written informed consent was obtained from every individual. A compensation of 8.44 USD was given to each study participants as time compensation. The participants identified with impaired hearing loss were referred for further evaluation and management. Data with study identification numbers only was recorded in the Wulira App. It is stored on a password protected closed based server.

## Results

We recruited 400 potential participants from a pool of 900 industrial workers. Participants who did not work in the noisy sections of the industry were excluded (n = 500). Of the 400 eligible participants, twenty (n = 20) of them declined to participate in the study, ten (n = 10) had bilateral outer ear pathologies while 16 were excluded because the target sample size was achieved (Fig 2).

### Socio-demographic and clinical characteristics

Of the 354 participants, 333 (94.1%) were male, and the median age was 27, IQR, 25–30. Of 354 participants, 161 (45.4%) were machine operators (Fig 3).

Nearly half of the participants (159, 44.9%) had completed Ordinary level education, while one hundred thirty-seven (38.7%) were protestant. The median estimated monthly income was 88.9 USD, IQR, (72.2–133.3) USD. 148 (41.8%) were single while majority (334, 94.3%) of the participants resided outside Kampala district. Regarding the risk factors of hearing loss, fourteen (3.9%) had history of smoking and more than half (65.5%) had worked in the industry for more than 2 years. Nine (2.5%) reported history of MDR- TB treatment (Table 1).

### Prevalence of OHL among industrial workers

The overall prevalence of OHL among industrial workers was 11.3% (40/354). However, the perceived prevalence of hearing loss was 5.7% (20/354) in the left ear and 5.9% (21/354) in the right ear. The average prevalence of hearing loss was 63 (17.8%) in the right ear and 38 (10.7%) in the left ear.

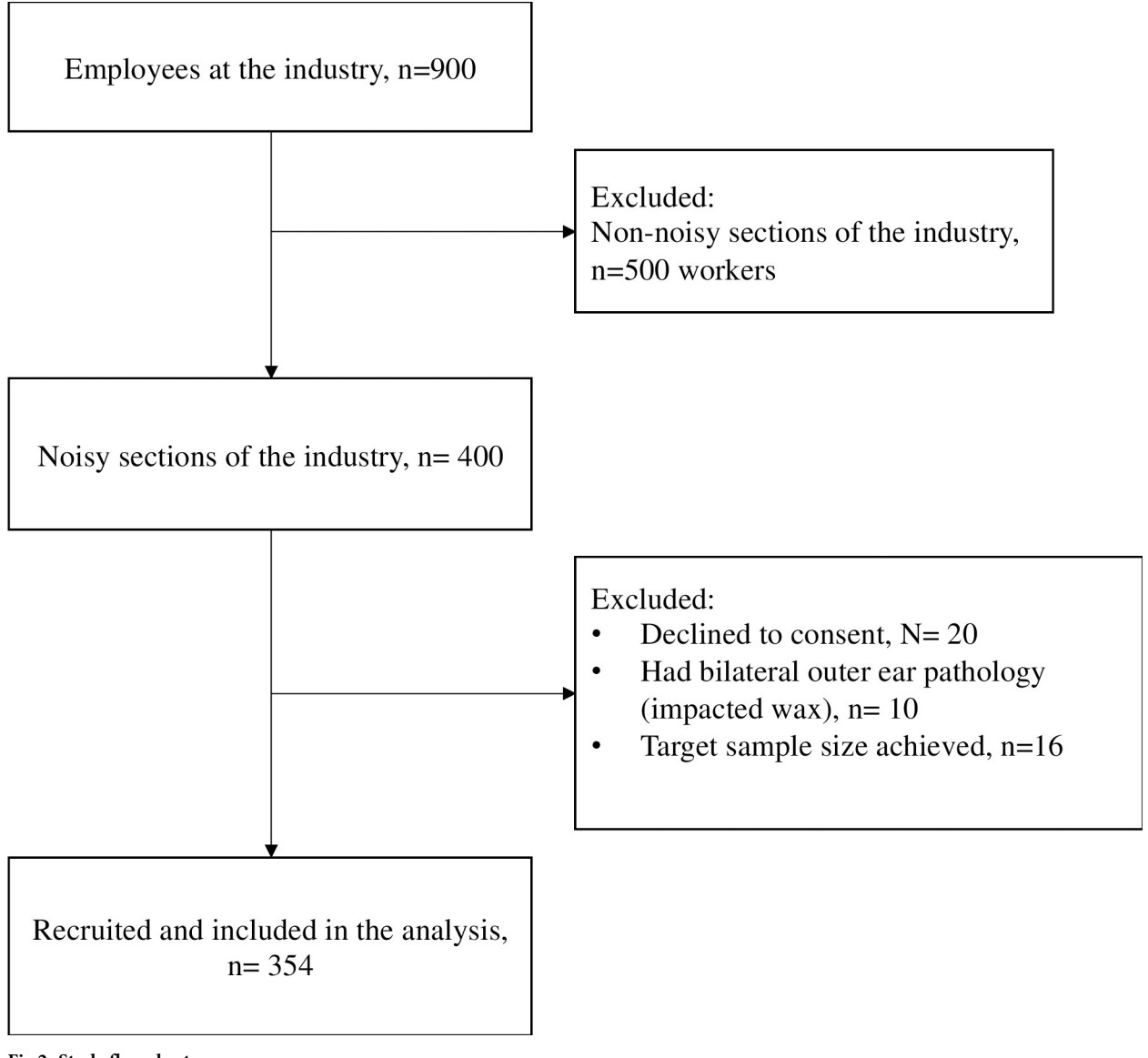

**Fig 2. Study flow chart.**

On further categorization of hearing loss, 57 (16.1%) and 32 (9%) had mild hearing loss in the right and left ear respectively (Table 2).

As shown in Table 3, more than quarter of the participants had hearing loss in the left ear (109, 30.8%) and right ear (116, 32.8%) at 250 Hz. Approximately one third of the participants, 116 (32.8%) and 129 (36.4%) had low frequency hearing loss (<2000 Hz) in the left and right ear respectively. Few individuals had high frequency hearing loss in our study (51 (14.4%)-left ear versus 73 (20.6%) -right ear). Fig 4 shows a comparison between the prevalence of perceived hearing loss and objectively assessed hearing loss.

**Audiometric notch**

Less than five percent, 14 (4%) of the participants had notch in their right ear while 7 (2%) had notch in their left ear. According to the duration of work, participants who had worked for

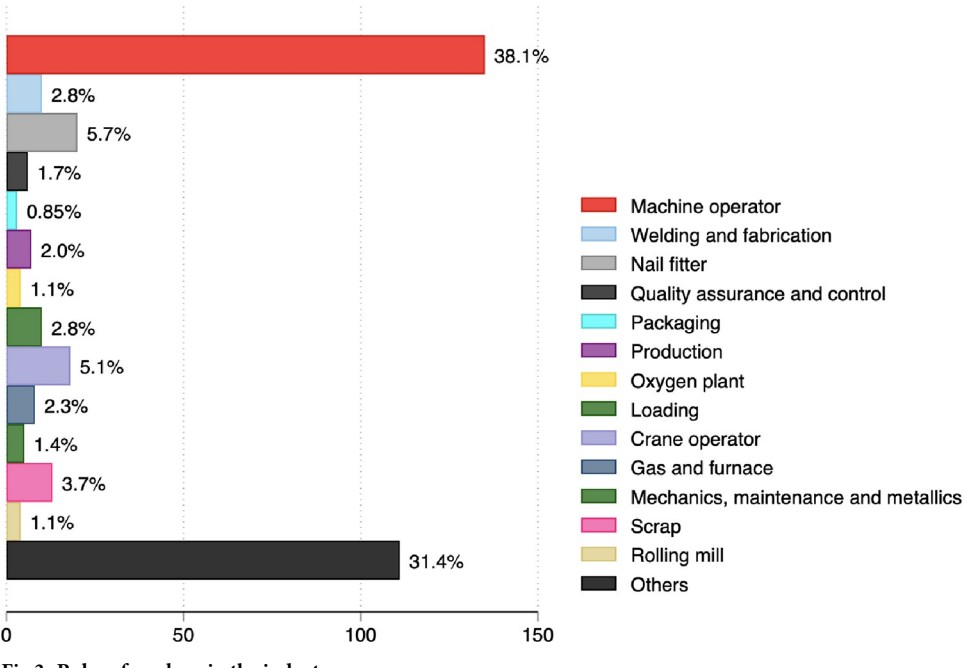

**Fig 3. Roles of workers in the industry.**

more than 24 months had the highest prevalence of notch in the right and left ear at 8 (3.5%) and 3 (1.3%) respectively (Table 4).

## Factors associated with hearing loss

Residing outside Kampala district was associated with hearing loss (Odds ratio, 95% Confidence Interval, 0.213 (0.063–0.725), p = 0.013) (Table 5). However, no other factor assessed including age, gender, level of education, marital status, duration of working at the factory, and history of previous treatment for MDR-TB was found to be associated with hearing loss.

## Discussion

Our cross-sectional study aimed at determining the prevalence of OHL among industrial workers. We found that the prevalence of OHL was 11.3%. The average prevalence of hearing loss was 10.7% in the left ear and 17.8% in the right ear. The prevalence of low frequency OHL was 32.8% in the left ear and 36.4% in the right ear while high frequency hearing loss was 14.5% in the left ear and 20.6% in the right ear respectively.

Although the prevalence of OHL in our study falls within the global prevalence of OHL which ranges from 11.2% to 58% [40], it is less than the regional prevalence of 17%- 48% [3, 6, 8, 20]. For instance, a study in Tanzania among industrial workers reported a prevalence of 46.5% and 50.8% in area A and B respectively [41] while another study among 253 steel and iron industrial workers reported a prevalence of 48% [3] The lower prevalence reported in our study may be attributed to the use of Wulira App to screen for OHL while other studies used PTA, the gold standard to assess for OHL. The protective gear and noise reduction measures set up by the formal industrial sector could have further contributed to the low prevalence. Age is an important risk factors of hearing loss [42] especially age-related hearing loss [43]. In this study, we found that participants were younger than those found in other studies. The participants in our study may have had a shorter duration of exposure to harmful noise as well as

**Table 1. Socio-demographic and clinical characteristics of the study population.**

| Characteristic | Frequency, N (%) |
|---|---|
| **Sex** | |
| Female | 21 (5.9) |
| Male | 333 (94.1) |
| **Age in completed years** | |
| Median (Inter quartile Range (IQR)) | 27 (5) |
| <25 | 83 (23.5) |
| 25–35 | 234 (66.1) |
| >35 | 37 (10.4) |
| **Roles in the industry** | |
| Machine operators | 161 (45.4) |
| Welding and fabrication | 18 (5.1) |
| Nail fitting | 20 (5.7) |
| Quality assurance and control | 6 (1.7) |
| Packaging and Loading | 33 (9.3) |
| Mechanics, maintenance and metallics | 5 (1.4) |
| Others | 111 (31.4) |
| **Level of education** | |
| No formal education | 1(0.3) |
| Primary education | 31 (8.8) |
| Ordinary Level Education | 159 (44.9) |
| Advanced Level of Education | 65 (18.4) |
| University | 33 (9.3) |
| Other tertiary institution | 65 (18.4) |
| **Marital status** | |
| Married | 92 (26.0) |
| Divorced | 5 (1.4) |
| Single | 148 (41.8) |
| Cohabiting | 107 (30.2) |
| Others | 1 (0.6) |
| **Estimated monthly income in USD*** | |
| Median (IQR) | 88.9 (61.1) |
| **Residence** | |
| Kampala district | 20 (5.7) |
| Not in Kampala district | 334 (94.3) |
| **Clinical characteristics** | |
| **How long have you worked here** | |
| Less than 6 months | 28 (7.9) |
| 6–12 months | 28 (7.9) |
| 13 to24 months | 66 (18.6) |
| More than 24 months | 232 (65.5) |
| **Have you ever smoked** | |
| No | 340 (96.1) |
| Yes | 14 (3.9) |
| **Have been treated for multi drug resistant tuberculosis** | |
| No | 345 (97.5) |
| Yes | 9 (2.5) |

*1 USD = 3600, IQR- Inter-quartile range.

**Table 2. Occupational hearing loss among industrial workers.**

| Prevalence of OHL | | |
| --- | --- | --- |
| | **Left ear** | **Right ear** |
| **OHL category** | | |
| Normal (< = 25dB) | 316 (89.3) | 291 (82.2) |
| Hearing loss (>25 dB) | 38 (10.7) | 63 (17.8) |
| **Hearing loss sub-categorization based on WHO classification** | | |
| **Hearing loss category** | **Left ear** | **Right ear** |
| Normal (< = 25dB) | 316 (89.3) | 291 (82.2) |
| Mild loss (26–40 dB) | 32 (9.0) | 57 (16.1) |
| Moderate loss (41–60 dB) | 4 (1.1) | 4 (1.1) |
| Severe and profound loss (61 and above) | 2 (0.6) | 2 (0.6) |

highly unlike to have comorbidities such as hypertension or prior exposure to ototoxic chemicals that could cause damage to the hair cells. This may partly explain the lower prevalence reported.

OHL was predominant in the right ear with the highest number of participants having mild hearing loss which differs from other settings [44, 45]. Even though this was an unexpected finding among workers exposed to noise from different directions/sources simultaneously, published literature has noted asymmetrical hearing loss is common among individuals exposed to chronic noise [46, 47]. The asymmetry in our study could be attributed to the method used to assess for OHL and the stage of OHL. The Wulira App has been shown to have a low sensitivity to mild, moderate, or severe hearing loss [30] which could have limited the application's ability to diagnose OHL. We suggested that the application should be used as a screening tool in the normal population. In addition, our study could not readily establish the difference in pattern hence the need for further casual studies in our population. Like in

**Table 3. OHL at different frequencies.**

| Frequency | Left ear, N (%) | Right ear, N (%) |
| --- | --- | --- |
| **250Hz** | | |
| Normal (< = 25dB) | 245(69.2) | 238 (67.2) |
| Abnormal (>25dB) | 109 (30.8) | 116 (32.8) |
| **500 Hz** | | |
| Normal (< = 25dB) | 308 (87.0) | 301 (85.0) |
| Abnormal (>25dB) | 46 (13.0) | 53 (15.0) |
| **1000 Hz** | | |
| Normal (< = 25dB) | 344 (97.2) | 332 (93.8) |
| Abnormal (>25dB) | 10 (2.8) | 22(6.2) |
| **2000 Hz** | | |
| Normal (< = 25dB) | 344 (97.2) | 338 (95.5) |
| Abnormal (>25dB) | 10 (2.8) | 16 (4.5) |
| **4000 Hz** | | |
| Normal (< = 25dB) | 331 (93.5) | 310 (87.6) |
| Abnormal (>25dB) | 23 (6.5) | 44 (12.4) |
| **8000 Hz** | | |
| Normal (< = 25dB) | 323 (91.2) | 313 (88.4) |
| Abnormal (>25dB) | 31 (8.8) | 41 (11.6) |

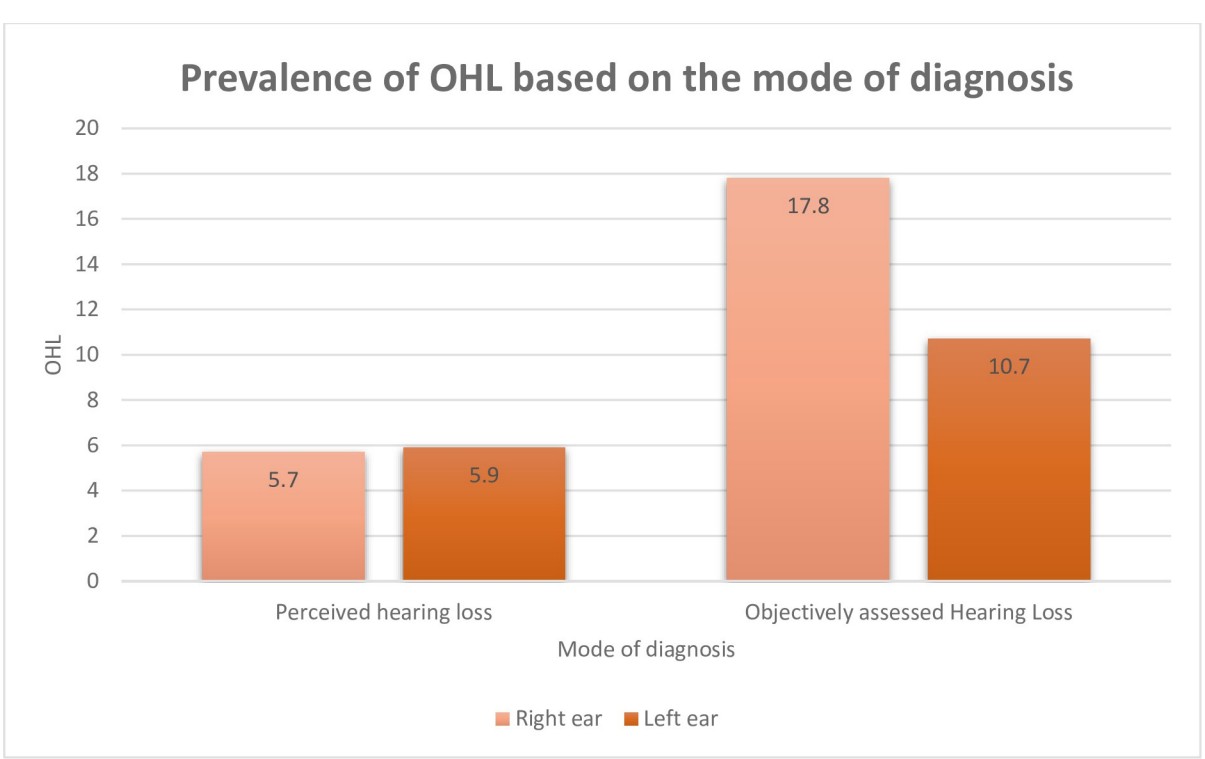

**Fig 4. A bar graph showing a comparison between the prevalence of perceived hearing loss and objectively assessed hearing loss.**

Hafiz et al. study [48], we found that perceived hearing loss was lower than objectively assessed hearing loss. This mismatch highlights the need to regularly screen for hearing loss among industrial workers.

Our study revealed that industrial workers have low frequency hearing loss which differs from a study in China which is a industrialized country [49]. Continuous exposure to excessive

**Table 4. Audiometric notch among the participants.**

| Characteristic | Frequency | Percentage | | | |
|---|---|---|---|---|---|
| Left ear notch present | | | | | |
| No | 347 | 98.0 | | | |
| Yes | 7 | 2.0 | | | |
| Right ear notch present | | | | | |
| No | 340 | 96.0 | | | |
| Yes | 14 | 4.0 | | | |
| **Prevalence of notch based on duration of working at the industry.** | | | | | |
| Characteristic | <6 months, n (%) | 6–12 months, N (%) | 13–24 months, N (%) | >24 months, N (%) | P value |
| Left ear notch present | | | | | |
| No | 27 (96.4) | 27 (96.4) | 64 (97.0) | 229 (98.7) | |
| Yes | 1 (3.6) | 1 (3.6) | 2 (3.0) | 3 (1.3) | 0.290* |
| Right ear notch present | | | | | |
| No | 27 (96.4) | 27 (96.4) | 62 (93.9) | 224 (96.5) | |
| Yes | 1 (3.6) | 1 (3.6) | 4 (6.1) | 8 (3.5) | 0.778* |

*P value is based on Fisher's exact test.

**Table 5. Regression analysis for factors associated with hearing loss.**

| Characteristic | Odds ratios | 95% Confidence interval | p-value | Significant |
|---|---|---|---|---|
| **Education level** | | | . | |
| Primary level or no formal education | 1 | | . | |
| secondary | 1.145 | 0.236–5.555 | 0.867 | |
| university or tertiary | 1.195 | 0.222–6.436 | 0.836 | |
| **Age** | | | . | |
| <25 years | 1 | | . | |
| 25–35 years | 0.83 | 0.259–2.658 | 0.753 | |
| above 35 years | 2.249 | 0.52–9.729 | 0.278 | |
| **Sex** | | . | . | |
| Female | 1 | | | |
| Male | 2.213 | 0.264–18.527 | 0.464 | |
| **Marital status** | | . | . | |
| Married/cohabiting | 1 | . | . | |
| divorced/divorced | 0.438 | 0.155–1.241 | 0.12 | |
| **Duration of working at the factory** | | . | . | |
| Less than 6 months | 1 | . | . | |
| 6 months to 1 year | 0.249 | 0.024–2.529 | 0.24 | |
| More than 1 years | 0.641 | 0.155–2.655 | 0.539 | |
| More than 2 years | 0.393 | 0.114–1.356 | 0.14 | |
| **Residence of the participant** | | . | | |
| In Kampala | 1 | | | |
| Not in Kampala | 0.213 | 0.063–0.725 | 0.013 | ** |
| **History of previous treatment for MDR TB*** | | | | |
| No | 1 | | | |
| Yes | 1.645 | 0.181–14.979 | 0.659 | |

*MDR-TB-Multi Drug resistant Tuberculosis.

loud sound could lead to temporary threshold shift and/ or permanent threshold shift consequently causing impairment in the transmission of low or high frequency sounds [50, 51]. This was an unexpected finding which may be because of our inability to perform bone conduction thresholds, ambient noise level measurements of the industrial environment as well as performing an immitansmetric examination before the audiometric measurement. Furthermore, it could be attributed to inherent performance of Wulira app performance to assess for hearing loss which might have led to the low estimation of the prevalence of high frequency hearing loss. Although we were not able to document the ambient noise levels at our study site, studies have reported high noise levels industrial settings [9, 17–20, 41, 52]. Based on these factors, we hypothesize that the high prevalence of low frequency hearing loss among our study population could be attributed to the high noise levels. This calls for further studies to determine the ambient noise levels of industrial settings in Uganda and establish its association with hearing loss at different frequency levels.

Less than five of participants had audiometric notch in the left and right ear respectively. Moreover, the highest number of participants who had audiometric notch had worked in the industry for more than 2 years. Compared to a study among Norwegian railway workers who were exposed to noise [53], our study had a lower prevalence of notch using Wilson's definition [39]. Additionally, bilateral presence of notch has been reported in a study done among male industrial workers [28]. Although the presence of notch was high in participant who had

worked in the industry for more than 2 years, duration of noise exposure was statistically insignificant among iron factory male workers in Taiwan [28]. Similarly, the association of audiometric notch with duration of work or noise exposure in the industry was not statistically significant. The cross-sectional nature of our study may not provide the exact estimate of the causal relationship between audiometric notch and duration of work which provides an avenue for further longitudinal studies to be conducted. Audiometric notch has been suggested as one of the characteristics of OHL [54] and is crucial in differentiating between noise induced hearing loss and age associated hearing loss [27]. However, the occurrence of notch corroborates our finding that hearing loss in this population was most likely resulting from the nature of their work, which could be exposes to excessive noise or ototoxic chemicals.

Residing away from Kampala district was associated with hearing loss in our study population. This could possibly be to the high level of exposure to ototoxic agents such solvents, heat, or recreational noise in these areas.

The study was done among formal industrial workers employed in a well-established industry; therefore, our findings may not be applicable to other settings. We were unable to measure the ambient noise in the different sections of the factory where the study participants worked. Although this may have predisposed individuals to varying noise levels, our study recruited individuals who were considered exposed to the noisy sections of the factory. In our assessment of hearing loss, we did not consider the 6kH which may minimally bias our conclusions on the OHL. We did not assess for pre-existing underlying diseases/conditions that are known to cause low frequency hearing loss like Meniere's disease in our population. Even with these limitations, our study sample size was large and we use a clinically validated mobile phone application which offers hearing assessment similar to the PTA [30]. Additionally, whereas PTA measures frequencies of up to 4000 Hz, Wulira App measures frequencies above 4000 Hz up to 8000 Hz and can therefore, be used to screen for hearing loss at higher frequencies. In addition, Wulira App is readily accessible and cost effective when compared to PTA. This study provides valuable evidence to guide development of occupation health policies and frameworks. Additionally, it provides a baseline for longitudinal studies to evaluate the effectiveness of Wulira app in continuously screen for OHL and understanding the changes in OHL in this population.

## Conclusion

In this study, OHL was highly prevalent among industrial workers when screened using mobile audiometry, *Wulira app*. There is need to strengthen noise reduction measures in industries. To aid the implementation of the mandatory accurate periodic screening of hearing among industrial workers, mobile audiometry should be scale up to strengthen preventive measures.

## Acknowledgments

We are grateful to the participants and administrative staff of industry who granted us permission and took part in the study.

## Author Contributions

**Conceptualization:** Charles Batte, Andrew Weil Semulimi, John Mukisa.

**Data curation:** Charles Batte, Andrew Weil Semulimi, John Mukisa.

**Formal analysis:** Charles Batte, Andrew Weil Semulimi.

**Funding acquisition:** Charles Batte, Immaculate Atukunda.

**Investigation:** Charles Batte, Andrew Weil Semulimi, John Mukisa.

**Methodology:** Charles Batte, Immaculate Atukunda, Andrew Weil Semulimi, Mariam Nakabuye, Nelson Twinamasiko, John Mukisa.

**Project administration:** Charles Batte, Immaculate Atukunda, Mariam Nakabuye.

**Resources:** Joab Mumbere.

**Software:** Joab Mumbere.

**Supervision:** Charles Batte, Andrew Weil Semulimi, John Mukisa.

**Validation:** Charles Batte, Andrew Weil Semulimi, Mariam Nakabuye, Festo Bwambale, Joab Mumbere, David Mukunya, Israel Paul Nyarubeli, John Mukisa.

**Visualization:** Andrew Weil Semulimi, Mariam Nakabuye, Festo Bwambale, Joab Mumbere, Nelson Twinamasiko, David Mukunya, Israel Paul Nyarubeli, John Mukisa.

**Writing – original draft:** Andrew Weil Semulimi.

**Writing – review & editing:** Charles Batte, Immaculate Atukunda, Andrew Weil Semulimi, Mariam Nakabuye, Festo Bwambale, Joab Mumbere, Nelson Twinamasiko, David Mukunya, Israel Paul Nyarubeli, John Mukisa.

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
