## [Decision Letter · Decision Letter 0]

1 Sep 2022

PONE-D-22-16420Using mobile audiometry (Wulira app) to assess noise induced hearing loss among industrial workers in Kampala, Uganda: A Cross-sectional studyPLOS ONE

Dear Dr. Semulimi,

Thank you for submitting your manuscript to PLOS ONE. After careful consideration, we feel that it has merit but does not fully meet PLOS ONE’s publication criteria as it currently stands. Therefore, we invite you to submit a revised version of the manuscript that addresses the points raised during the review process.

We look forward to receiving your revised manuscript.

Kind regards,

Paul Hinckley Delano, Ph.D.

Academic Editor

PLOS ONE

Journal Requirements:

“This study was funded by the Government of Uganda through the Research and Innovation Fund Makerere University, Fund MAKRIF/ DVCFA/ 026/ 20. The content is solely the responsibility of the authors and does not necessarily represent the official views of the Research and Innovation Fund.”

“Dr. Charles Batte is part of the team that developed the Wulira App. Dr. Charles Batte, Joab Mumbere, and Dr. Andrew Weil Semulimi serve as employees of Wulira Health Limited in which Wulira App is a subsidiary. The other authors have no conflict of interest to declare.”

5. We noted in your submission details that a portion of your manuscript may have been presented or published elsewhere. [DETAILS AS NEEDED] Please clarify whether this [conference proceeding or publication] was peer-reviewed and formally published. If this work was previously peer-reviewed and published, in the cover letter please provide the reason that this work does not constitute dual publication and should be included in the current manuscript.

Additional Editor Comments:

Reviewer were diverse. In order to be considered for publication the authors should give more analysis and discussion about differences between left and right ears. Take into account all the comentaries raised by reviewers. Include an example figure.

Reviewers' comments:

Reviewer's Responses to Questions

**Comments to the Author**

1. Is the manuscript technically sound, and do the data support the conclusions?

Reviewer #1: Yes

Reviewer #2: No

Reviewer #3: Yes

2. Has the statistical analysis been performed appropriately and rigorously? 

Reviewer #1: Yes

Reviewer #2: No

Reviewer #3: Yes

3. Have the authors made all data underlying the findings in their manuscript fully available?

Reviewer #1: Yes

Reviewer #2: Yes

Reviewer #3: Yes

4. Is the manuscript presented in an intelligible fashion and written in standard English?

Reviewer #1: Yes

Reviewer #2: Yes

Reviewer #3: Yes

5. Review Comments to the Author

Reviewer #1: It is a very good study but I have a doubt. The industry had 900 employees, 300 were excluded for not working exposed to noise. That gives me 600 and not 400 employees exposed to noise. Is it a mistake?

Reviewer #2: There is a difference in the rate of hearing loss between ears. Was this significant? The authors include this fact in the discussion, nevertheless, this an unexpected finding in workers exposed to noise coming from several sources simultaneously.

There was a difference between low-frequency and high-frequency hearing loss, the former being more prevalent. This is also an unexpected finding since occupational hearing loss is characterized by high-frequency hearing loss.

The last two comments lead me to check the original publication about the Wulira app. The paper previously published by the authors showed that the app had a good sensitivity and specificity to detect subjects with normal hearing. But, the sensitivity dropped to approximately 53% when subjects had mild hearing loss, and even below 50% for more severe cases. Maybe the app should be employed as a screening tool for identifying subjects with normal hearing levels, but the use as a diagnostic method of levels of hearing loss is still questionable. This issue could explain the findings of differences between ears and the higher prevalence of low-frequency hearing loss.

This is the reason I do not recommend this paper for publication

Reviewer #3: The paper study the prevalence of occupational hearing loss among workers of metal manufactoring company located in Kampala, Uganda. The assesment was performed using the Wulira App. It was found that about 10 percent of the workers of this company had occupational hearing loss. The results highlight the importance of hearing loss screening among this populations and the usefulness of the Wulira App as an alternative to pure tone audiometry as a screening tool.

The paper is well written and clear. The methodology is sound and the results are interesting. For these reasons, I recommend accepting the paper after the following issues are addressed:

* Major Issues

- Show some examples of the audiometries obtained with the Wulira App. For instance, show one audiometry of a subject with occupational hearing loss and one of a subject without it. This will allow the reader to get a better understanding of the kind hearing loss observed in the population studied.

- Use a bar plot for figure 2. See https://en.wikipedia.org/wiki/Pie_chart#Use_and_effectiveness for the rationale about avoiding pie charts. I suggest checking this reference (in particular section 6.1) for some guidelines about appropriately presenting bar graphs: https://clauswilke.com/dataviz .

- Line 202: Explain why reporting the religion is relevant. And if it is, please also report the other religions of the population.

- In the paragraph starting at line 239 you compare the prevalence of your results with other studies, and suggests some reasons to explain the differences. One factor that you do not consider to explain the difference is the use of the Wulira App. Is it possible that differences in the results are because you use a different tool to get the audiogram? Please comment about this issue in the discussion section.

* Minor Issues

- Line 68: Change "Occupational hearing loss" by "Occupational Hearing Loss."

- Line 91: The sentence "Much as the Constitution ..." is hard to read. Please re-write to facilitate the understanding.

- Line 94: Add a coma before and after of "which is the gold standard for hearing loss screening."

- Line 97: Add a coma before and after "which is hearing loss screening using mobile phones and tablets."

- Line 197: Change "achieved. (Figure 1)" by "achieved (Figure 1)."

- Line 225: Change "respectively. (Table 4)" by "respectively (Table 4)."

6. PLOS authors have the option to publish the peer review history of their article (what does this mean?). If published, this will include your full peer review and any attached files.

Reviewer #1: **Yes: **Carolina Der

Reviewer #2: **Yes: **Mariela C. Torrente

Reviewer #3: **Yes: **Alejandro Weinstein

---

## [Author Response · Author response to Decision Letter 0]

7 Oct 2022

Reviewer #1: It is a very good study, but I have a doubt. The industry had 900 employees, 300 were excluded for not working exposed to noise. That gives me 600 and not 400 employees exposed to noise. Is it a mistake?

Response: Thank you for your comment. Yes, this is mistake in the write up. The actual number exposed to noise is 400 as outlined in the study population section of the manuscript. The actual number of employees excluded is 500. The tables have been updated as such.

Reviewer #2: There is a difference in the rate of hearing loss between ears. Was this significant? The authors include this fact in the discussion, nevertheless, this an unexpected finding in workers exposed to noise coming from several sources simultaneously. There was a difference between low-frequency and high-frequency hearing loss, the former being more prevalent. This is also an unexpected finding since occupational hearing loss is characterized by high-frequency hearing loss.

The last two comments lead me to check the original publication about the Wulira app. The paper previously published by the authors showed that the app had a good sensitivity and specificity to detect subjects with normal hearing. But the sensitivity dropped to approximately 53% when subjects had mild hearing loss, and even below 50% for more severe cases. Maybe the app should be employed as a screening tool for identifying subjects with normal hearing levels, but the use as a diagnostic method of levels of hearing loss is still questionable. This issue could explain the findings of differences between ears and the higher prevalence of low-frequency hearing loss.

This is the reason I do not recommend this paper for publication

Response: Thank you for your comment. Yes, there is a statistically significant difference between the hearing loss between the ears. The P value is less than 0.001. 

It is true that this is an unexpected finding among workers exposed to noise from different directions/sources simultaneously. Previous research has, however, noted differences in hearing loss by ear side among Finnish individuals exposed to chronic noise (1) while other researchers have demonstrated asymmetry in OHL (2).This has been added to the discussion lines 253—259.

Regarding the high prevalence of low frequency noise induced hearing loss, this contrasts with previous studies as we have highlighted in our discussion previously. We also agree that the frequency of “high frequency hearing loss” should be more for noise induced hearing loss than that seen in our study. The other possible explanation for this could be the inherent Wulira app performance as you highlight. This has been added to the discussion, line 270—271. The other underlying diseases/conditions for causing low frequency hearing loss like Meniere’s disease in our population were not ruled and this has been added as a limitation, line 303—304. 

We have also highlighted throughout the manuscript that the tool should be used as a screening tool.

Reviewer #3: The paper study the prevalence of occupational hearing loss among workers of metal manufacturing company located in Kampala, Uganda. The assessment was performed using the Wulira App. It was found that about 10 percent of the workers of this company had occupational hearing loss. The results highlight the importance of hearing loss screening among this populations and the usefulness of the Wulira App as an alternative to pure tone audiometry as a screening tool.

The paper is well written and clear. The methodology is sound, and the results are interesting. For these reasons, I recommend accepting the paper after the following issues are addressed:

* Major Issues

- Show some examples of the audiometry obtained with the Wulira App. For instance, show one audiometry of a subject with occupational hearing loss and one of a subject without it. This will allow the reader to get a better understanding of the kind hearing loss observed in the population studied.

Response: Thank you for your comment. These have been added

- Use a bar plot for figure 2. See https://en.wikipedia.org/wiki/Pie_chart#Use_and_effectiveness for the rationale about avoiding pie charts. I suggest checking this reference (in particular section 6.1) for some guidelines about appropriately presenting bar graphs: https://clauswilke.com/dataviz .

Response: Thank you for your comment. A bar plot has been added

- Line 202: Explain why reporting the religion is relevant. And if it is, please also report the other religions of the population.

Response: Thank you for your comment. Religion is not important and hence has been removed.

- In the paragraph starting at line 239 you compare the prevalence of your results with other studies and suggests some reasons to explain the differences. One factor that you do not consider explaining the difference is the use of the Wulira App. Is it possible that differences in the results are because you use a different tool to get the audiogram? Please comment about this issue in the discussion section.

Response: Thank you for your comment. We included a sentence explaining the fact that other studies used PTA as compared to ours that utilized Wulira App. See lines 243—244.

* Minor Issues

- Line 68: Change "Occupational hearing loss" by "Occupational Hearing Loss."

Response: Thank you for your comment. This have been addressed. 

- Line 91: The sentence "Much as the Constitution ..." is hard to read. Please re-write to facilitate the understanding.

Response: Thank you for your comment. This have been addressed.

- Line 94: Add a coma before and after of "which is the gold standard for hearing loss screening."

Response: Thank you for your comment. This have been addressed.

- Line 97: Add a coma before and after "which is hearing loss screening using mobile phones and tablets."

Response: Thank you for your comment. This have been addressed.

- Line 197: Change "achieved. (Figure 1)" by "achieved (Figure 1)."

Response: Thank you for your comment. This have been addressed.

- Line 225: Change "respectively. (Table 4)" by "respectively (Table 4)."

Response: Thank you for your comment. This have been addressed.

References.

1. Pirilä T, Sorri M, Jounio-Ervasti K, Sipilä P, Karjalainen H. Hearing asymmetry among occupationally noise-exposed men and women under 60 years of age. Scandinavian Audiology. 1991;20(4):217-22.

2. Le TN, Straatman LV, Lea J, Westerberg B. Current insights in noise-induced hearing loss: a literature review of the underlying mechanism, pathophysiology, asymmetry, and management options. Journal of Otolaryngology-Head & Neck Surgery. 2017;46(1):1-15.

---

## [Decision Letter · Decision Letter 1]

8 Dec 2022

Using mobile audiometry (Wulira app) to assess noise induced hearing loss among industrial workers in Kampala, Uganda: A Cross-sectional study

PONE-D-22-16420R1

Dear Dr. Semulimi,

We’re pleased to inform you that your manuscript has been judged scientifically suitable for publication and will be formally accepted for publication once it meets all outstanding technical requirements.

Kind regards,

Paul Hinckley Delano, Ph.D.

Academic Editor

PLOS ONE

Additional Editor Comments (optional):

Reviewers' comments:

Reviewer's Responses to Questions

**Comments to the Author**

1. If the authors have adequately addressed your comments raised in a previous round of review and you feel that this manuscript is now acceptable for publication, you may indicate that here to bypass the “Comments to the Author” section, enter your conflict of interest statement in the “Confidential to Editor” section, and submit your "Accept" recommendation.

Reviewer #3: All comments have been addressed

Reviewer #4: All comments have been addressed

2. Is the manuscript technically sound, and do the data support the conclusions?

Reviewer #3: Yes

Reviewer #4: Partly

3. Has the statistical analysis been performed appropriately and rigorously? 

Reviewer #3: Yes

Reviewer #4: Yes

4. Have the authors made all data underlying the findings in their manuscript fully available?

Reviewer #3: Yes

Reviewer #4: Yes

5. Is the manuscript presented in an intelligible fashion and written in standard English?

Reviewer #3: Yes

Reviewer #4: Yes

6. Review Comments to the Author

Reviewer #3: (No Response)

Reviewer #4: The article presented a study of the prevalence of occupational hearing loss among workers in a metal industry company in Kampala district. All recommendations have been addressed by the authors in the current version of the manuscript.

7. PLOS authors have the option to publish the peer review history of their article (what does this mean?). If published, this will include your full peer review and any attached files.

Reviewer #3: **Yes: **Alejandro Weinstein

Reviewer #4: No

---

## [Editor Report · Acceptance letter]

23 Dec 2022

PONE-D-22-16420R1 

Using mobile audiometry (Wulira app) to assess noise induced hearing loss among industrial workers in Kampala, Uganda: A Cross-sectional study 

Dear Dr. Semulimi:

I'm pleased to inform you that your manuscript has been deemed suitable for publication in PLOS ONE. Congratulations! Your manuscript is now with our production department. 

Kind regards, 

on behalf of

Dr. Paul Hinckley Delano 

Academic Editor

PLOS ONE